# Opposite Roles of BAP1 in Overall Survival of Uveal Melanoma and Cutaneous Melanoma

**DOI:** 10.3390/jcm9020411

**Published:** 2020-02-03

**Authors:** Feng Liu-Smith, Yunxia Lu

**Affiliations:** 1Department of Epidemiology, University of California Irvine, Irvine, CA 92697, USA; 2Department of Medicine, University of California Irvine, Irvine, CA 92697, USA; 3Chao Family Comprehensive Cancer Center, University of California Irvine, Irvine, CA 92697, USA; yunxial1@uci.edu; 4Department of Population Health and Disease Prevention, University of California Irvine, Irvine, CA 92697, USA

**Keywords:** uveal melanoma, cutaneous melanoma, BAP1, overall survival, age difference

## Abstract

Background: BRCA1-Associated Protein 1 (BAP1) germline mutations predispose individuals to cancers, including uveal melanoma (UM) and cutaneous melanoma (CM). BAP1 loss is common in UM and is associated with a worse prognosis. BAP1 loss is rare in CM and the outcome is unclear. Methods: UM and CM data was retrieved from The Cancer Genome Atlas (TCGA) database. Cox regression model was performed to examine whether BAP1 mRNA levels or copy number variations were associated with overall survival (OS). Results: BAP1-low mRNA predicted a poor OS in UM (HR = 9.57, 95% CI: 2.82, 32.5) but a contrasting better OS in CM (HR = 0.73, 95% CI: 0.56, 0.95). These results remained unchanged after adjusting for sex, age, and stage in UM and CM, or after adjusting for ulceration or Breslow depth in CM. Additionally, low BAP1 mRNA predicted a better OS in CM patients older than 50 years but not in younger patients. Co-expression and enrichment analysis revealed differential genes and mutations that were correlated with BAP1 expression levels in UM and CM tumors. Conclusions: Low BAP1 mRNA was significantly associated with a better OS in CM patients, in sharp contrast to UM. High BAP1 expression in CM was significantly associated with over-expressed CDK1, BCL2, and KIT at the protein level which may explain the poor OS in this sub-group of patients. Function of BAP1 was largely different in CM and UM despite of a small subset of shared co-expressed genes.

## 1. Introduction

Germline mutations in BAP1 (BRCA1-Associated Protein 1) are associated with multiple types of hereditary cancers [1], which is now classified as BAP1-TPDS (BAP1 Tumor Predisposition Syndrome) [2]. The susceptible cancer types include malignant mesothelioma [3], lung adenocarcinoma, meningioma [1], gallbladder cancer [4], renal cell carcinomas [5], cutaneous melanoma (CM) [6], uveal melanoma (UM) [1,7] and myelodysplastic syndrome [8]. The BAP1 germline mutations are generally deletions or loss-of-function mutations, thus BAP1 was defined as a tumor suppressor gene [8]. About 31% of BAP1-TPDS individuals were reported to develop UM [9]. Somatic mutations in UM tumors are also common. In an analysis of germline vs. somatic mutations in UM tumors, 8% of tumors were reported to have germline mutation while 43–45% of tumors had somatic mutations [10,11]. Overall, the germline mutations in BAP1 are rare in CM (<1% of CM tumors carry BAP1 germline mutations) [12]. Additionally, tumor characteristics of the BAP1-TPDS-associated CM are quite unique, exhibiting distinct morphology and histology which are similar to Atypical Spitz Tumors [2], which are substantially different from other common CM sub-types. 

BAP1 is a deubiquitinase that inhibits cell growth in Hela cells and breast cancer cells [13,14], and promotes DNA damage-associated apoptosis, thus reduces cellular transformation [15]. Loss of BAP1 in uveal melanoma cells did not impact cell proliferation or tumorigenesis; rather, loss of BAP1 led to de-differentiation accompanied by increased stem-like biomarkers [16]. In cutaneous melanoma cell lines, however, stable over-expression of BAP1 promoted cell growth while knockdown of BAP1 suppressed cell proliferation with concomitant decrease of survivin, an anti-apoptotic protein [17]. Therefore, the role of BAP1 in cell growth and tumorigenesis seems to be context-dependent. 

BAP1 mutations are associated with worse prognosis and survival in UM patients due to increased metastatic potentials [11,18]. A meta-analysis of various cancer types with BAP1 mutations indicated that BAP1 mutations were also associated with worse outcomes in renal cell carcinoma but not in other cancers [19]. High BAP1 mRNA expression levels were reported to be associated with a better survival in a cohort of primary CM patients [17], which was consistent with BAP1 function as a tumor suppressor in CM, but the authors stated that BAP1 level was perhaps confounded by other prognosis factors such as ulceration and Breslow depth in that study [17]. 

## 2. Experimental Section

Data Source: This is a secondary data analysis based on TCGA (The Cancer Genome Atlas) sequencing and patient information, which were retrieved from the GDC (Genomics Data Commons) data portal (https://portal.gdc.cancer.gov/) [20,21]. A total of 469 CM patients (TCGA-SKCM, Skin Cutaneous Melanoma) and 80 UM patients (TCGA-UVM, Uveal Melanoma) were included for analysis. As all personal information was de-identified and not trackable, the Institutional Review Board (IRB) at the University of California Irvine (UCI) determined that this study was a non-human subject study and therefore did not require a specific IRB protocol. Tumor staging followed AJCC (American Joint Committee on Cancer) TNM (The Tumor, Node, Metastasis) criteria using pathological evidence. Characteristics in CM included tumor TNM stages (AJCC pathological), presence or absence of ulceration, Breslow depth, age of diagnosis, and sex of patients, which were used in multivariate Cox regression analysis. The original publication and data were from Robertson et al. for UM and Alexandrov et al. for CM [22,23]. The TCGA CM and UM patient information were level 1 and 2 raw data; mRNA expression data were RSEM-normalized (RNA-Seq by Expectation Maximization) [24]; mutations (single nucleotide changes or small insertion/deletions) and copy number variations (CNVs) in each tumor were level 3 processed data.

Survival curves for mRNA expression levels of BAP1 or BAP1 copy number variation (CNV) were plotted using Kaplan–Meier method. The hazard ratio (HR) and 95% confidence interval of BAP1 CNV status or expression levels (low versus high) were first analyzed by univariate Cox regression model, with Breslow method to handle tied failures. Influence of other prognostic factors, such as age of diagnosis, sex of the patients, and stage of disease were analyzed using multi-variate Cox analysis. The Cox proportional hazard assumption was tested using the Schoenfeld residuals method and no variable violated the assumption. Comparison of means was analyzed by Student’s t-test, analysis of variance, or linear regression. Association of BAP1 levels with age was analyzed by linear regression models, with or without adjustment to stage or sex. The genomic level of enrichment and co-expression analysis on BAP1 was performed using the cBioportal-embedded MEMo software [21,25]. Statistical significance was set at 0.05 and adjusted by Bonferroni method for multiple comparison. With the estimated HR of 9.0 for UM, and 0.7 for CM, the sample size required for UM and CM was 7 and 247, respectively, in order to reach 80% of statistical power [26]. Therefore, 80 UM patients and 469 CM patients were sufficient for the statistical power, although the total numbers were imbalanced between the two melanoma types. All statistical analysis was performed using Stata (IC 13.1) software (College Station, TX, USA) unless otherwise specified. 

## 3. Results

### 3.1. Low BAP1 mRNA or Loss of BAP1 Indicated Significant Worse Survival in UM Patients

The UM tumors were divided into low and high mRNA levels using 50% cut off values or in tertiles using the downloaded mRNA RSEM data. As shown in Figure 1a, the UM patients with low mRNA expression showed significant worse overall survival as compared to those with higher mRNA expression, regardless of the cut-off value at 50% or in tertiles (Figure 1a,b). When the cut-off value was set at 50%, the HR was 9.57 (95% confidence interval (CI): 2.82, 32.5, *p* < 0.0001) (Figure 1a, Table 1). When BAP1 mRNA was grouped into tertiles, the global statistics *p* value was 0.0011, with the middle tertile exhibiting HR of 8.3 (95% CI: 1.04, 65.6) and lower tertile exhibiting HR of 14.5 (95% CI: 1.9, 110.6), as compared to the top tertile which had the highest BAP1 mRNA expression (Table 1). Thus, lower levels of mRNA in UM indicated significantly worse overall survival, which was consistent with the role of BAP1 as a tumor suppressor in UM [11]. 

The mRNA expression may not be influenced by mutation status. For example, with a loss-of-function point mutation in the exon, the gene may show high levels of mRNA but the expressed protein may not be functional. In this case, high level of mRNA expression may become a misclassification in the survival analysis. In order to address this issue, the correlation of mutation status, CNV and mRNA expression levels were examined in the UM cohort. Among the 80 UM tumors, 13 tumors carried a BAP1 point mutation and 44 showed copy number variation (CNV = −1, all hemizygous deletion, defined as BAP1-CNV group). All 13 BAP1-mutated tumors had hemizygous deletion, and therefore were included in the 44 BAP1-CNV category. Hemizygous deletion may indicate monosomy 3, which was a driver cause for uveal melanoma [18,22]. Although the mean mRNA levels were significantly lower in BAP1-CNV tumors (Appendix A), five BAP1-CNV tumors were categorized in the top half mRNA group and one diploid BAP1 in the lower half group. CNV does not always correlate with mRNA expression levels [27], therefore, a Cox regression analysis was performed based on the BAP1 CNV status. The results were similar to those obtained from the mRNA analysis, with the BAP1-CNV group exhibiting significantly worse overall survival than the non-altered group (HR = 13.0, 95% CI: 3.0, 56.0) (Figure 1c).

### 3.2. Multivariate Cox Regression in UM Patients

Next, multivariate Cox regression models were used to analyze the impact of age, sex, and stage on BAP1-associated overall survival. Univariate analysis indicated that age, but not sex or stage, was significantly associated overall survival in the UM group (*p* = 0.019 for age of diagnosis, 0.33 for sex, and 0.10 for stage). All these three factors were used to adjust the BAP1 effect; the BAP1 effect remained significant after adjusting to age of diagnosis (HR 9.03, 95% CI: 2.67, 30.6), sex (HR = 9.59, 95% CI: 2.83, 32.47), or stage of tumor (HR 8.85, 95% CI: 2.57, 30.51) (Table 2). BAP1 mRNA levels remained a significant factor in the fully adjusted model with all three variables included (Table 2).

### 3.3. Low BAP1 mRNA Indicated a Significant Better Survival in CM patients

In contrast to the observations in the UM patient cohort, CM patients with low BAP1 mRNA expression showed significant better overall survival as compared to patients with high BAP1 mRNA expression, either with 50% cutoff or in tertiles (Figure 1d,e, Table 1). HR was 0.73 (95% CI 0.56, 0.95) in the 50% cutoff model (*p* = 0.019). HR was 0.73 (95% CI: 0.53, 1.02) for the second tier, and 0.67 (95% CI: 0.48, 0.93) for the third tier (lowest mRNA) in the tertile model (Table 1). 

Eleven tumors carried BAP1 point mutations, with four silent synonymous mutations and seven missense mutations with unknown significance (I643T, E30K, P629S, R417M, S143N (*N* = 2), L416F and R59W). It was not surprising that these mutations were not associated with overall survival in CM (HR = 0.58, *p* = 0.36). Of the 373 tumors having BAP1 CNV information, 74 had hemizygous deletion, 227 had no change, and 66 had amplification. The copy numbers of the BAP1 gene correlated with mRNA expression levels in CM (Appendix A), i.e., tumors with BAP1 hemizygous deletion also showed lower BAP1 mRNA RSEM estimates. Compared to patients with no BAP1 copy number alteration (i.e., BAP1 diploids), patients with BAP1 amplification or deletion both showed significant better survival (Table 1, Figure 1f). BAP1 deletion indicated non-significant better survival (HR = 0.74, *p* = 0.116); and BAP1 amplification (correlated with high level of mRNA) indicated significant better survival (HR = 0.56, *p* = 0.005) (Table 1). The proportional hazards assumption was not violated (*p* = 0.26). BAP1 amplification-predicted better survival remained significant after adjusting to age of diagnosis (HR 0.59, *p* = 0.013), but the significance was lost after adjusting to stage of disease (HR = 0.68, *p* = 0.08) (Appendix A). BAP1 amplified tumors had slightly lower percentage (12/55 = 21.8 vs. 42/178 = 23.6%) of stage T1 and slightly higher percentage of stage T4 disease (14/55 = 25.5% vs. 39/178 = 21.9%) as compared to diploid tumors. 

The follow-up time in the UM and CM cohorts were quite different. The longest follow-up time for UM patients were 84.5 months, while that for CM patients was 370.0 months. The median follow-up was 25.8 and 36.7 months for UM and CM, respectively. Although the Cox proportional hazards models and Kaplan–Meier curves both account for the discrepancy in follow-up time, we still performed an analysis to investigate whether the results held true if we shorten the follow-up time to 84.5 months in the CM patients. In order to do this, we re-defined “death” event as “living” if the follow-up time was over 84.5 months in the CM cohort; Cox proportional hazard model was then used to analyze the new “overall” survival. As shown in Appendix A, lower BAP1 mRNA levels remained as a significant predictor for better “overall” survival with HR of 0.73 and *p* value of 0.041. 

### 3.4. Multivariate Cox Regression Analysis in CM Patients

In order to examine whether BAP1 expression levels were confounded by other prognostic factors, multivariate Cox regressions were performed. First univariate regression was used to assess the significance of each known variable in overall survival. Sex did not show significant influence (*p* = 0.35), but stage of disease, age of diagnosis, presence of ulceration, and Breslow depth were all significant (all *p* values < 0.001). Each of these variables were used to adjust for BAP1 effect (50% cutoff model) in the multivariate Cox model. The results are shown in Table 2. Lower BAP1 mRNA levels remained a significant indicator for better overall survival after adjusting to stage of disease, ulceration, and Breslow depth in the regression analysis (Table 2). BAP1 mRNA remained a significant indicator in the fully adjusted model which included age, stage, and sex (HR = 0.73, 95% CI: 0.54, 0.99, Table 2). However, the significance of BAP1 mRNA was lost when the overall survival was adjusted to only age of diagnosis, regardless whether age was used as a continuous variable or as a categorial variable (*p* = 0.099 and 0.076, respectively) (Table 2). 

Hence, we investigated whether BAP1 mRNA levels were correlated with age of diagnosis. Linear regression analysis showed that BAP1 RSEM level in CM was significantly correlated with age of diagnosis beta = 15.8, *p* < 0.001) (Figure 2a). However, the R-squared value was low (0.045). When the patients were grouped into young (≤50 years) and old (>50 years) groups, the mean for the older group (3030.0 ± 68.7) was significantly higher than that from the younger group (2640.7 ± 74.7, *p* <0.0001, Student *t*-test) (Table 3). In comparison, BAP1 mRNA showed a non-significant decreasing trend with age in the UM patient cohort (*p* = 0.16, linear regression) (Figure 2b). More interestingly, when a Cox analysis was performed in different age strata (≤50 and >50 years), BAP1 mRNA was a significant predictor for survival, only in the older age group but not in the young age group (Table 3). Of course, age itself is a significant predictor for overall survival (Appendix A). 

### 3.5. Differential and Shared Molecular Networks of the BAP1 in CM and UM

In order to understand the potential mechanisms of BAP1 function in UM and CM, differential molecular networks of the BAP1 in each tumor were analyzed using the cBioportal-embedded MEMo software [25]. Co-expressed genes were analyzed in the entire cohort for CM and UM, respectively. Using cutoff values of Pearson′s correlation coefficient of 0.4 and adjusted *p* < 10^−6^, excluding genes located on Chromosome 3p (where BAP1 is located), a total of 233 negatively correlated and 84 positively correlated genes exhibited significant co-expression with BAP1 in CM tumors. In UM patients, a total of 1563 genes (719 positively correlated and 844 negatively correlated) showed significant co-expression with BAP1 mRNA using the same cutoff standards as in CM. Thirty-nine of the co-expressed genes were shared by UM and CM (Appendix A), all of which exhibited the same direction of regulation (positive or negative correlation). One of these genes is PMEL, which encodes a melanosome structural protein shared by both CM and UM. The majority co-expressed genes were unique for UM or CM. Top 20 co-expressed genes (ranked by Pearson’s coefficient, in either direction) in UM and CM are listed in Appendix A. There were no shared genes in this subset. 

Next, we asked whether specific mutations or CNVs were associated with high BAP1 expression in CM or low BAP1 expression in UM, each of which indicated worse overall survivals. A Z score of −1.4 was used for OQL query (Onco Query Language [21]) in UM, which defined 50% (*n* = 40) of samples as low BAP1 expression. Enrichment analysis (embedded in cBioportal, [21]) revealed that EIF1AX (unadjusted *p* = 0.0072) and SF3B1 (unadjusted *p* = 0.0072) mutations were exclusive with low expression of BAP1. Mutation rates of UM oncogenes GNAQ and GNA11 were similar in BAP1-low tumors as compared to the rest of tumors. Exclusivity of SF3B1 and EIF1AX mutations with BAP1 was consistent with previous reports [11,28], and supported classification of UM in three sub-types by mutations in three genes: BAP1, SF3B1, and EIF1AX [28]. SF3B1 CNV was not common in the UM cohort, with one homozygous deletion, and eight hemizygous amplifications out of the 80 tumors. SF3B1 mRNA levels were not associated with overall survival (HR = 1.29, *p* = 0.66, bottom 30% vs. top 30%); but SF3B1 mutation alone or mutation plus amplification indicated better overall survival (Appendix A). This outcome was perhaps due to the presence of SF3B1 mutation/amplification exclusively in the BAP1-high group. Neither EIF1AX mRNA levels nor the mutation status was associated with UM overall survival. 

For the CM cohort, a cutoff Z score of > −0.17 represented the top half of BAP1 mRNA tumors (*n* = 231 vs. the other half *n* = 232) and was used in the OQL query [21]. No significant mutations or CNVs were associated with high BAP1 mRNA expression. The 9p21-22 deletion (including CDKN2A, CDKN2B, MTAP, RPS6, ACER2, and RN7SKP258 genes) showed mutual exclusivity with BAP1 high mRNA (*p* = 0.01 ~0.0003, not significant after multiple comparison adjustment). When the RPPA (Reverse Phase Protein Array) profile was examined, pro-proliferation proteins, such as KIT, BCL1, and CDK1 were significantly over-expressed in the BAP1-high tumors, while the apoptotic protein CASP7 was under-expressed (Appendix A).

## 4. Discussion

This study described opposite roles of BAP1 in survival of two types of melanoma patients. The worse outcome of BAP1-low patients (including hemizygous deletion of BAP1 and low BAP1 mRNA) was validated in UM patients, while a new discovery of low BAP1 mRNA indicating a better overall survival in CM patients was described. Furthermore, low BAP1 mRNA seemed to indicate a significantly better survival for CM patients of older age (>50 years) but did not predict survival for younger patients. The age-differentiated BAP1 role in CM survival is different than that in UM where loss of BAP1 predicted worse outcome in patients of all ages [18]. UM and CM diagnosis was usually at younger ages in the BAP1-TPDS group as compared to the general population [10,29,30], but UM tumors carrying germline mutations required a longer time to progress to metastasis than those carrying somatic mutations [10]. Our previous publications suggested that there was an age- and sex-differentiated etiology in CM [31,32]; discovering age-associated prognosis markers may be helpful in revealing the age-differentiated tumor causes. 

These results are consistent with the cellular role of BAP1 in CM cell lines A375 and C918 where depletion of BAP1 expression led to an inhibition of cell growth [17] but were different with the survival outcome in the same Kumar et al. study [17], which showed low BAP1 levels, indicating worse outcomes in CM. The difference may be because CM cases were all primary tumors in the Kumar et al. study but the TCGA cohort contained a whole spectrum of tumor stages including metastatic tumors. BAP1 mRNA levels did not show a difference among different stages in our analysis and the stage-adjusted model also showed low BAP1 mRNA indicating significantly better survival in CM. Thus, it is unknown what caused the discrepancy in these results. It is also a mystery why BAP1 amplification was associated with better overall survival, even though BAP1 amplification was correlated with higher BAP1 mRNA levels. Future analysis based on larger datasets and more confounding factors may be required to completely elucidate the function of BAP1 in CM.

Molecular network analysis revealed that BAP1 was associated with very different molecular profiles in CM and UM. Specifically, low BAP1 expression or loss of BAP1 in UM was exclusive to SF3B1 or EIF1AX mutations, while high BAP1 expression in CM was non-significantly exclusive to 9p21 deletion. Although both melanomas produce melanin, their oncogenic pathways and mutation spectra are different. The primary oncogenic mutations in UM are GNAQ and GNA11, while that in CM are BRAF, NRAS, and PTEN. Nevertheless, BAP1 expression levels were not associated with GNAQ and GNA11 mutation status in UM tumors, nor with NRAS or PTEN mutation status in CM tumor. BAP1 mRNA did show significantly higher expression levels in BRAF-wild type CM tumors than in the BRAF-mutant tumors (3073.6 ± 76.0 vs. 2697.5 ± 72.8, *p* = 0.0006). When the BRAF mutation status was used for adjustment in the multivariate Cox model, low BAP1 mRNA levels were still significantly associated with a better OS (HR = 0.74, *p* = 0.026) while BRAF mutation status did not predict OS in either simple or multivariate Cox model. However, among the co-expressed genes, the melanosome structure protein PMEL was positively correlated with BAP1 mRNA levels in both tumors (#21 in Appendix A). An enrichment analysis using the RPPA protein database revealed that high BAP1 mRNA in CM was correlated with over-expression of proliferative protein KIT, CDK1, and BCL2, and under-expression of apoptotic protein CASP7. These results may explain why BAP1-high CM tumors exhibited poorer survival. Further understanding of BAP1 network regulation in these melanomas may provide opportunities for future therapy. 

One caveat of our analysis is that focusing on DNA or RNA aberrations may miss other mechanisms that cause loss of BAP1 protein function, such as loss of BAP1-interacting proteins. For example, BAP1-induced apoptosis in neuroblastoma cells is mediated via an interaction with the 14-3-3 protein [33]; thus, the loss of the 14-3-3 protein may lead to loss of BAP1 function (at least partially), even when BAP1 mRNA and DNA are normally expressed. The other possible caveat of this study is that the mutation and CNV sequence analyses did not capture the large complex re-arrangements of BAP1 locus, which would require an additional complex bioinformatics algorithm. However, it was reported that these additional structural variants were all associated with low BAP1 mRNA expression [22], therefore mRNA expression levels should not mis-classify these additional mutations.

## 5. Conclusions

BAP1 plays distinctively different roles in the overall survival of UM and CM patients. It is a new finding that low BAP1 mRNA levels were significantly associated with a better overall survival in CM patients, particularly in older patients. These results may reflect how distinct oncogenic signals impact BAP1 function in these two types of melanomas. Further investigation on cell context and oncogene-dependent function of BAP1 may provide molecular explanations of the observed epidemiological data.

## Figures and Tables

**Figure 1 jcm-09-00411-f001:**
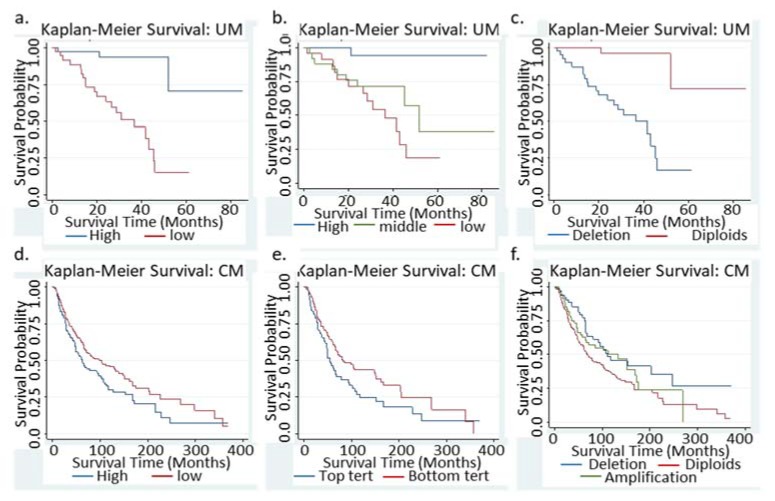
Kaplan–Meier survival curves in UM and CM. (**a**,**b**) Survival curves of patients with BAP1-low vs. BAP1-high tumors (mRNA level, bottom 50% vs. top 50% in (**a**), tertiles in (**b**). (**c**) Survival curves of UM patients with BAP1 copy number variation. (**d**,**e**) Survival curves of BAP1-low vs. BAP1-high tumors in CM patients. (**d**) BAP1-tumors with bottom 30% mRNA or with BAP1 mutations. These two categories contained all BAP1 hemizygous deletion tumors in UM. (**f**) survival curves of CM patients with various BAP1 copy number variations.

**Figure 2 jcm-09-00411-f002:**
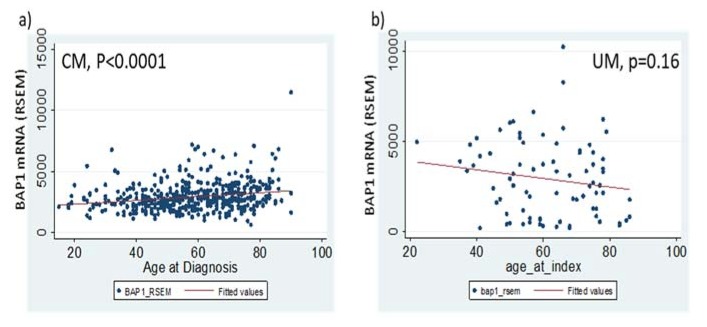
The trend of BAP1 mRNA expression along age axis in CM (**a**) and UM (**b**). Scatter plots of BAP1 RSEM (RNA-Seq by Expectation Maximization) estimates in each tumor with age of diagnosis as X axis. CM tumors were grouped by median mRNA Z scores (± 2.0) and screened for BAP1 co-expressed genes.

**Table 1 jcm-09-00411-t001:** BRCA1-Associated Protein 1 (BAP1) mRNA or copy number variation (CNV) predicted overall survival of uveal melanoma (UM) and cutaneous melanoma (CM) patients using the univariate Cox Regression model.

		UM					CM				
Model	Variables	N	HR	95% CI		*p*	N	HR	95% CI		*p*
50% Cutoff	High mRNA	40	ref	ref	ref		235	ref	ref	ref	
	Low mRNA	40	9.57	2.82	32.48	0.010	234	0.73	0.56	0.95	0.019
mRNA in Tertiles	Highest	26	ref	ref	ref	ref	158	ref	ref	ref	
	Middle	28	8.27	1.04	65.56	0.045	155	0.73	0.53	1.02	0.063
	Lowest	26	14.46	1.89	110.64	0.010	156	0.67	0.48	0.93	0.017
BAP1-CNV	No alteration	36	ref	ref	ref		227	ref	ref	ref	
	Hemi. Dele.	44	12.97	3.00	55.99	0.001	74	0.74	0.51	1.08	0.116
	amplification	0	N/A	N/A	N/A	N/A	66	0.56	0.37	0.84	0.005

Hemi. Dele: Hemizygous deletion, also referring to BAP1-CNV group in UM.

**Table 2 jcm-09-00411-t002:** Multivariate Cox models to predict BAP1 mRNA levels in UM and CM overall survival.

	UM					CM					Additional Variable(s)
Model	HR	95% CI		pBAP1	p_av *	HR	95% CI		pBAP1	p_av *	
1	9.03	2.7	30.6	<0.001	0.028	0.80	0.61	1.04	0.099	<0.001	Age (continuous)
2	9.60	2.8	32.60	<0.001	0.509	0.78	0.60	1.03	0.076	<0.001	Age as category **
3	8.85	2.6	30.5	0.001	0.565	0.70	0.53	0.94	0.019	<0.001	Stage ***
4	9.59	2.3	32.5	<0.001	0.299	0.73	0.56	0.95	0.019	0.396	Sex
5						0.71	0.52	0.99	0.042	<0.001	Ulceration
6						0.62	0.46	0.84	0.002	<0.001	Breslow Depth
7	10.13	2.96	34.63	<0.0001		0.73	0.54	0.99	0.042		Age, sex, stage

* p_av: *p* value for additional variables listed on the right side of the table; ** age was grouped based on ≤50 or >50 years at diagnosis. *** Stage: Based on American Joint Committee on Cancer (AJCC) pathological TNM staging, with T1 and T2 grouped as early stage while T3 and T4 as late stage. Model 7 represents the fully adjusted model including age, sex, and stage of disease in both UM and CM.

**Table 3 jcm-09-00411-t003:** Correlation of age with BAP1 mRNA levels in CM and age-specific survival analysis.

		*t*-Test, *p* = 0.0004		Cox Survival Analysis			
Age Group	N	Mean	Std. Err	HR	95% CI		p_BAP1
≤50	142	2640.1	75.7	1.00	0.59	1.68	0.995
>50	326	3034.1	69.6	0.71	0.51	0.98	0.047
All	468	2914.5	85.1	0.73	0.56	0.95	0.019

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
