# Peer review of "Opposite Roles of BAP1 in Overall Survival of Uveal Melanoma and Cutaneous Melanoma"

_jcm, 2020, doi:10.3390/jcm9020411_

Round 1

Reviewer 1 Report

This manuscript describes a bioinformatic analysis to investigate BAP1 expression levels in uveal and cutaneous melanoma. Sequencing data and patient information were retrieved from the Cancer Genome Atlas.

The overall analysis is adequate. The statistical analysis is pertinent. Appropriate figures and tables are included in the manuscript.

It is noteworthy that the results indicate a different role of BAP1 in the two diseases. Although several cancers have been included in the BAP1 Tumor Predisposition Syndrome, recent data seems to indicate that BAP1 may play a more complex role than a tumor suppressor gene.

Minor:

The discussion section should be revised because sometimes it is not clear.

Author Response

Thank you for your help! We have modified some original sentences in an attempt to clarify some unclear discussions.  We also added extra contents to make up for some previously ignored issues.  The changes are tracking-marked in the R1 version and the clean text is shown in the R1clean version.  

Reviewer 2 Report

Dear Editor,

thank you for trusting me with the review of this paper. It elaborates on an interesting subject, at least for skin and uveal melanoma experts. The research has been well conducted and mostly clearly presented. There are some issues I would like the authors to address: First, the imbalance between the number of uveal melanoma and skin melanoma patients included. They should provide some motivation for this, and add an explanation as to why this will not be problematic out of a statistical perspective. Second, the authors seem to not pay enough attention to the fact that loss of BAP1 protein function can be caused by other mechanisms than aberrations on the DNA or RNA level. If they only analyse their samples on the RNA level, these mechanisms will be missed and they will fail to understand their own results fully. Third, some editing of grammatics and the extensive use of several spaces between sentences will be needed. Below are som further specific comments. Thank you again!

Best regards

Gustav Stålhammar

Author Response

Thank you, Dr. Gustav Stålhammar, for your positive review and for your constructive suggestions.  

For the imbalance of the numbers of two cohorts, we performed additional statistical power calculation to show that both UM and CM cohorts have sufficient number of patients to achieve at least 80% of power (with α=0.05).  Therefore the discrepancy in the numbers should not affect our conclusion.  The power calculation is added to the "experimental section" (changes are marked in the R1 version).

For loss of BAP1 function through other mechanism: this is a very good point that we did not think about in our first version.  Indeed BAP1 function can be compromised by other mechanism such as loss of an interacting protein (for example, 14-3-3).  We have added a few sentences to address this issue in the discussion section (the last paragraph of discussion).

For grammar: we have read the manuscript and indeed found a few errors.  Thanks for pointing this out and we did our best to correct the most obvious errors (changes are marked in the R1 version).   

Reviewer 3 Report

Liu-Smith and Lu in the manuscript entitled “Opposite roles of BAP1 in overall survival of uveal melanoma and cutaneous melanoma” analyzed expression of the BAP1 gene in melanomas correlated to overall survival of uveal melanoma (UM) and cutaneous melanoma (CM) patients by examining TCGA database. The data suggested that BAP1 expression levels are covertly correlated to overall survival in UM comparing to CM patients.

Some concerns include:

Figure 1 used different time period to compare survival rates between UM patients (0-80 months) and CM patients (0-400 months). Is it better to be consistent? Authors compared BAP1 expression correlated with expression of some oncogenes and apoptotic genes in UM and CM and suggested that the expression patterns of these genes were similar in UM and CM although BAP1 expression was oppositely correlated to overall survival in the two types of melanomas. This reviewer wonders whether the authors may examine the distributions of mutations of p53, RB1, Pten, and Braf genes, which are important for the pathogenesis melanoma, in UM and CM and further determine the impact of relevant cancer drivers associated with BAP1 on the pathogenesis of UM and CM. Check the format and grammar.

Author Response

Thank you for helping us to improve our manuscript.  

For the inconsistency of the follow-up time in UM and CM patients, we added a paragraph in Section 3.3 to address this issue.  To put it simply, since both Cox model and Kaplan-Meier analysis methods directly deal with time-events, the difference in tracking time theoretically should not matter.  To be cautious about this issues, we performed additionally analysis on the basis of shortening the follow-up time to 84.5 months (which is the follow-up time for UM patients) for CM patients.  All CM patients followed more than 84.5 months were assumed "living status" regardless of actual living/death status.  BAP1-low patients still showed significant better overall survival under these conditions (Table S3).   

The gene expression patterns in UM and CM: We did not make it clear that the majority of co-expressed genes are in fact different in UM and CM although there were a small set of genes shared the similar expression pattern.  We added a sentence in the results section (Section 3.5) to emphasize this observation: "The majority co-expressed genes were unique for UM or CM."

Tumor suppressors and oncogene status (p53, RB1, Pten, and Braf genes) and their relationship with BAP1 expression:  This is an excellent questions.  The TP53 and RB pathways are less directly involved in UM and CM tumorigenesis as in most other cancer types, therefore we did not perform analysis based on the mutational status of these genes.   The most important oncogenes for UM are GNAQ and GNA11.  In our original version we compared BAP1 levels in GNAQ and GNA11 UM tumors and there was no correlation (Section 3.5).  The most important oncogenes and tumor suppressors in CM are BRAF, NRAS and PTEN.  We performed additional analysis and found that although NRAS and PTEN mutation status were not associated with BAP1 expression, BRAF-mutated CM tumors showed significant lower level of BAP1 mRNA.  However, the BRAF mutation status did not impact overall survival on its own or in the multivariate Cox analysis with BAP1.  These new findings are added in the discussion section.  

Grammar: we have read the manuscript and indeed found a few obvious errors.  Thanks for pointing this out and we did our best to correct the most obvious errors (changes are marked in the R1 version).